# Higher convergence of human-great ape enteric eukaryotic viromes in central African forest than in a European zoo: a One Health analysis

Victor Narat [1,10], Maud Salmona[2,3,10], Mamadou Kampo[4], Thibaut Heyer[5], Abdeljalil Senhaji Rachik[2], Severine Mercier-Delarue[2], Noémie Ranger[6], Stephanie Rupp[7], Philippe Ambata[8], Richard Njouom[9], François Simon[2,3], Jérôme Le Goff [2,3,11] ✉ & Tamara Giles-Vernick [4,11] ✉

Human-animal pathogenic transmissions threaten both human and animal health, and the processes catalyzing zoonotic spillover and spillback are complex. Prior field studies offer partial insight into these processes but overlook animal ecologies and human perceptions and practices facilitating human-animal contact. Conducted in Cameroon and a European zoo, this integrative study elucidates these processes, incorporating metagenomic, historical, anthropological and great ape ecological analyses, and real-time evaluation of human-great ape contact types and frequencies. We find more enteric eukaryotic virome sharing between Cameroonian humans and great apes than in the zoo, virome convergence between Cameroonian humans and gorillas, and adenovirus and enterovirus taxa as most frequently shared between Cameroonian humans and great apes. Together with physical contact from hunting, meat handling and fecal exposure, overlapping human cultivation and gorilla pillaging in forest gardens help explain these findings. Our multidisciplinary study identifies environmental co-use as a complementary mechanism for viral sharing.

Pathogenic sharing between human and wild animal populations constitute major threats to human and animal health[1,2]. The virus SARS-CoV-2 illustrates the worst consequences of anthropozoonosis (zoonotic spillover) and the risks associated with zooanthroponosis (spillback): SARS-CoV-2 apparently emerged from bat populations and probably infected other intermediate animal hosts before moving into human populations, causing significant morbidity and mortality; zooanthroponosis may create animal reservoirs that can generate new SARS-CoV2 variants[3–5]. Pathogenic spillback from humans into wild animals, moreover, can hamper conservation of protected species. Such multispecies risks may catalyze new pandemics in the future[6,7].

The processes leading to anthropozoonoses are complex, driven by genetic proximity between hosts, a pathogen's adaptive capacity, and human-animal contact, itself catalyzed by anthropogenic changes,

[1]Eco-anthropologie, MNHN/CNRS/Univ. Paris Cité, Paris, France. [2]Virology, AP-HP, Hôpital Saint Louis, Paris, France. [3]INSIGHT U976, INSERM, Université Paris Cité, Paris, France. [4]Anthropology and Ecology of Disease Emergence Unit, Institut Pasteur, Université Paris Cité, Paris, France. [5]Cermes3, Université Paris Cité, Paris, France. [6]Laboratoire de virologie, Institut fédératif de Biologie, Hôpital Purpan, CHU Toulouse, Toulouse, France. [7]Department of Anthropology, City University of NewYork – Lehman College, NewYork, NY, USA. [8]Ministry of Agriculture and Rural Development, Yaounde, Cameroon. [9]Centre Pasteur du Cameroun, Yaounde, Cameroon. [10]These authors contributed equally: Victor Narat, Maud Salmona. [11]These authors jointly supervised this work: Jérôme Le Goff, Tamara Giles-Vernick. ✉e-mail: jerome.le-goff@aphp.fr; tamara.giles-vernick@pasteur.fr

including demographic expansion, land use that fragments habitats, economic changes, wild animal trade, hunting and butchering[6,8,9]. Modeling studies help predict where, when, and who is at greatest risk for emerging anthropozoonoses[10]. Although less studied, zooanthroponoses appear to be facilitated by similar practices and processes[2]. Lacking, however, are fine-grained investigations of potential pathogen sharing between animals and people, and the ecologies, processes, and practices sustaining such sharing[11].

Existing evidence and analyses only partially illuminate the dynamics of zoonotic spillovers and spillbacks[12,13]. Many analyses focus on single pathogens, genera or families[14,15] and overlook granular evidence of variable human-animal interactions facilitating or mediating against pathogen sharing. Certain human-animal contact investigations elucidating cross-species interactions are weakened by imprecise definitions of contact types and exclusive attention to blood-borne transmission through hunting and butchering, neglecting other transmission modes[16,17]. Finally, investigations integrating host animal ecologies are rare.

This article bridges viral, ecological, and anthropological investigation to produce integrated insight into potential pathogen sharing and the complex interactions facilitating it. It compares the gut virome of humans, gorillas (*Gorilla gorilla*), and chimpanzees (*Pan troglodytes*) residing in two different sites (central African forest and European zoo), detailing their attendant socio-ecological systems[18] to explain how viral sharing may occur. We focus on human-great ape interactions because of the deep evolutionary relations between these primate species, long histories of shared pathogens[19], and frequent interactions between humans and great apes in forest and zoo settings. We analyze the gastrointestinal virome for two reasons: biological collections were noninvasive for protected great ape species, and stools contain important quantities of environmentally-persistent viruses that can facilitate indirect transmission.

We hypothesize that host species and environment will influence the intestinal virome, as intestinal bacteriome studies have shown through comparisons across several habitat types and between such habitats and zoos[20–24]. We also predict that in both sites, the human virome would more closely resemble that of chimpanzees because of phylogenetic proximity, and because of close daily physical and environmental contact, viral sharing between humans and both great apes would be greater in the zoo than in the Cameroonian forest.

The first and principal site of investigation, located in the southeastern Cameroonian dense rainforest, is home to rural people who derive their livelihoods from farming, gathering, hunting and fishing and who share forest and farming spaces with sympatric species of lowland gorillas (*Gorilla gorilla gorilla*) and chimpanzees (*Pan troglodytes troglodytes*). The second site, a European zoo with sympatric chimpanzees (*P.t. verus*) and gorillas (*G.g. gorilla*) and their human zookeepers, offered a comparative setting where physical and environmental contact is high, continuous, and easily observed and offers conditions that may facilitate viral sharing[25].

The present study compares across two sites the shared gastrointestinal eukaryotic virome between humans and two nonhuman primate (NHP) species living in close proximity. Our metagenomic sequencing of human and great ape intestinal virome, conducted identically across the two sites, illuminates shared, potentially pathogenic viruses[4]. The human virome contains less-explored viral communities that are associated with disease conditions, trigger immune response, or may function as commensals[26,27], and few published studies have explored great ape gastrointestinal eukaryotic virome, pathogenic viruses and disease[28–30]. Crucially, our multidisciplinary analysis identifies gorilla and human co-use of small forest gardens as a complementary mechanism for viral sharing, in addition to physical contact through great ape hunting, meat handling and fecal exposure.

We begin with southeastern Cameroonian perspectives on their understandings of their history with great apes and socio-cultural perceptions that in turn influence their environmental and physical contacts with gorillas and chimpanzees. We then document the degree of intestinal virome-sharing between humans and great apes and chimpanzee-gorilla virome differences. Using this One Health approach, we evaluate relative influences of host phylogeny and proximity, ecology, and human perceptions and practices to identify mechanisms for cross-species viral sharing (Fig. 1).

## Results

### Humans and great apes in Cameroon: a lengthy, shared history

Southeastern Cameroonians' conceptions of their lengthy, intimate history with great apes influence their current perceptions, interactions, and contacts with great apes in the forest. They still recount several mythical tales (*likano*) during evening festivities, portraying a distant past in which people, chimpanzees, gorillas, and gods once lived together in settlements. This cohabitation ruptured when great apes committed social transgressions, resulting in their ejection from human society (Likano sessions, 07.12.2015, 07.08.2015).

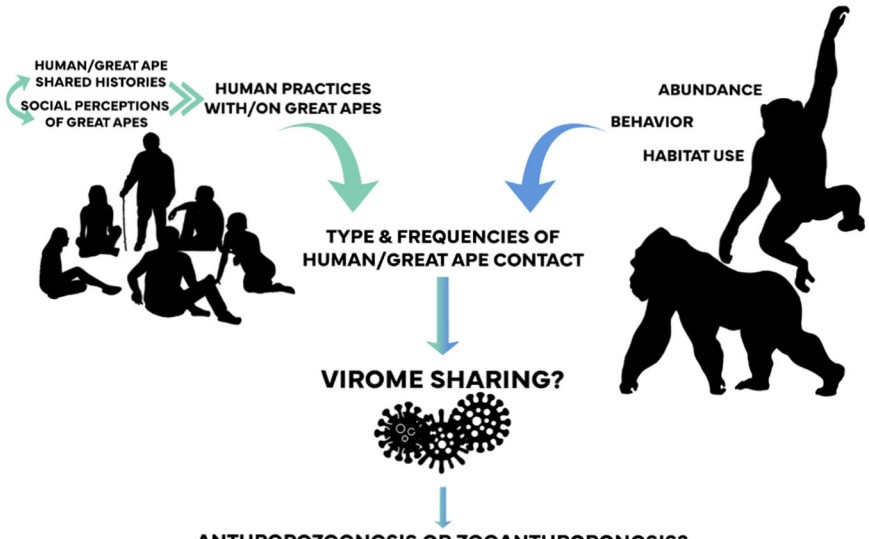

**Fig. 1 | Influences of human histories, perceptions, and practices and great ape ecology and behavior on eukaryotic intestinal virome sharing.** Graphic silhouettes come from Adobe Stock (Adobe for Enterprise version 5.10.0.573).

They contend that great apes continue these transgressions in the present, through crop raiding or harvesting too many wild fruits.

This deep historical cohabitation also underpins current local assumptions that people and great apes share specific capacities, notably emotional expressions of affection, sorrow, mutual protection, and actions such as mutual grooming. Great apes nonetheless displayed superior strength to humans, and local healers asserted that they used certain great ape body parts (tibia or vertebrae of chimpanzees, bone marrow from gorillas) useful in remedies for human weaknesses or ailments (Interviews 05.21.2016, 01.20.2016). The frequency of these uses is unknown, but they are not known to facilitate microbial sharing.

Mythical tales recounting past human-great ape cohabitation, ruptured by great apes' exile, followed a narrative structure mirrored in recent oral histories, reflecting local observations of a physical and emotional distancing between people and great apes accelerating over the last 50 years (Interviews 07.15.2015, 05.29.2017, 05.30.2017, 06.02.2017, 04.30.2016). As one aging hunter observed,

Before...gorillas slept next to the village. Now, they are there, but very rarely next to the village. They are afraid and very shrewd. Even the chimpanzees, there's a group between this and the next village... just a few meters from the road. But they don't cry out, they are prudent. It's rare to hear them, even though they are there (Interview 07.15.2015).

Whereas for some, intensified hunting had reduced gorilla and chimpanzee populations with the influx of high-powered rifles in the 1970s, for others, these great ape populations have remained stable over the long term (Interviews 01.21.2016, 05.13.2016, 05.14.2016, 05.29.2017, 05.30.2017, 06.02.2017).

These historical perspectives influenced human expectations of great ape transgressive behaviors (raiding) and of certain shared capacities of people and great apes, as well as their more recent emotional and physical distancing.

## Differential human contacts with chimpanzees and gorillas in Cameroon

These historical perspectives and expectations were further nuanced by differential perceptions of gorillas and chimpanzees, indirectly influencing the types and frequencies of human contacts with these NHPs. Southeastern Cameroonians did not recognize great apes as an NHP category; they distinguished chimpanzees (waké) from gorillas (ko), engaging with chimpanzees differently and less frequently than with gorillas.

Our data show that southeastern Cameroonians perceived chimpanzees as more intelligent than gorillas, capable of learning and displaying behaviors that approximate, but do not replicate, human behavior. Chimpanzees were more elusive than gorillas, living close to human settlements but remaining silent and invisible. Some specialist hunters also claimed that it was easier to kill a gorilla than a chimpanzee, contending that this relative ease partly resulted from gorillas' large size and practice of moving across the forest floor, rendering them more detectable than chimpanzees, who frequently traversed tree canopies (Interviews 05.31.2017, 01.18.2016). They also reported that emotionally, it was more difficult to kill a chimpanzee than a gorilla, because the former more closely resembled humans. As one former hunter confided, "You need a strong heart to kill a chimpanzee."

Southeastern Cameroonians reported that gorillas were less discerning, clumsier, more destructive, more frequently encountered than chimpanzees, and more unpredictably aggressive (Interviews 01.18.2016, 01.19.2016, 01.20.2016, 01.21.2016). Reportedly unable to distinguish ripe from unripe foods, gorillas more often laid waste to forest gardens, whereas chimpanzees purportedly raided only ripe foods from these gardens. Among people responding to a questionnaire conducted in four villages, 94% (424/449) experienced crop raiding by all NHPs, including gorillas and chimpanzees. Gorillas were more involved in the last event of crop raiding (50%, 225/449), contrary to chimpanzees (2%, 7/449). Moreover, 82% of respondents (366/449) reported that gorillas more often damaged gardens, and just 2% (7/449) cited chimpanzees as more frequent pillagers.

Quantitative data on human physical and environmental contacts, collected in real time in one village, also illuminates different human interactions with gorillas and chimpanzees. The mean frequency of human physical and environmental contact with gorillas was higher, but not significantly, than with chimpanzees; direct contact (seen alive, heard) did not differ between these great apes (Table 1). Based on questionnaires, physical contacts were more frequent (except for hunting) with gorillas than chimpanzees.

**Table 1 | Mean (%) contact frequencies (SD) according to great ape species and type of contact in Cameroon**

| Contact category | Type of contact | Longitudinal survey N = 18, data self-collected daily, 10 months | | | Questionnaire N = 449 | | |
|---|---|---|---|---|---|---|---|
| | | *Pan t. troglodytes* | *Gorilla g. gorilla* | Wilcoxon Test | *Pan t. troglodytes* | *Gorilla g. gorilla* | Wilcoxon Test |
| Environmental contact | Seen feces | 2.3 (5.3) | 3.5 (5.3) | NS | NA | NA | NA |
| | Seen food remains | 2.5 (5.3) | 4.0 (5.6) | NS | NA | NA | NA |
| | Seen nest | 1.9 (3.5) | 1.5 (2.3) | NS | NA | NA | NA |
| | Seen footprints | 2.2 (5.2) | 3.8 (5.3) | NS | NA | NA | NA |
| Direct contact | Seen alive | 1.8 (5.1) | 1.8 (4.1) | NS | 14.2 (31.4)[a] | 9.1[a] (22.3) | NS |
| | Heard | 3.2 (5.3) | 2.6 (4.2) | NS | 14.2[a] (31.4) | 9.1[a] (22.3) | NS |
| Physical contact | Hunt | 0 (0, 0-0) | 0 (0, 0-0) | NS | 0.08 (0.7) | 0.07 (0.3) | $P = 0.059$ $W = 103,997$ |
| | Butcher | 0.1 (0.3) | 0.2 (0.3) | NS | 0.7 (2.2) | 1.6 (7.3) | ***$P < 0.001$ $W = 116,996$*** |
| | Cook | 0.1 (0.4) | 0.2 (0.6) | NS | 0.6 (2.1) | 1.0 (2.7) | ***$P < 0.001$ $W = 119,923$*** |
| | Consume | 0.2 (0.5) | 1.1 (2.5) | NS | 0.7 (2.2) | 1.7 (7.3) | ***$P < 0.001$ $W = 118,773$*** |
| | Buy/Sell | 0.5 (1.2) | 1.1 (2.6) | NS | 0.4 (1.6) | 1.3 (7.1) | ***$P < 0.001$ $W = 115,929$*** |

Physical contact frequencies are from Narat et al. 2018. We compared the mean contact frequency, for each type of contact between humans and chimpanzees and between humans and gorillas, based on the longitudinal survey and questionnaire dataset with two-sided Mann–Whitney statistical tests. P values < 0.05 are in bold.
*NS* Not significant, *NA* Not addressed in questionnaires.
[a]In the questionnaires, direct contact was considered to be one contact type (seen alive or heard).

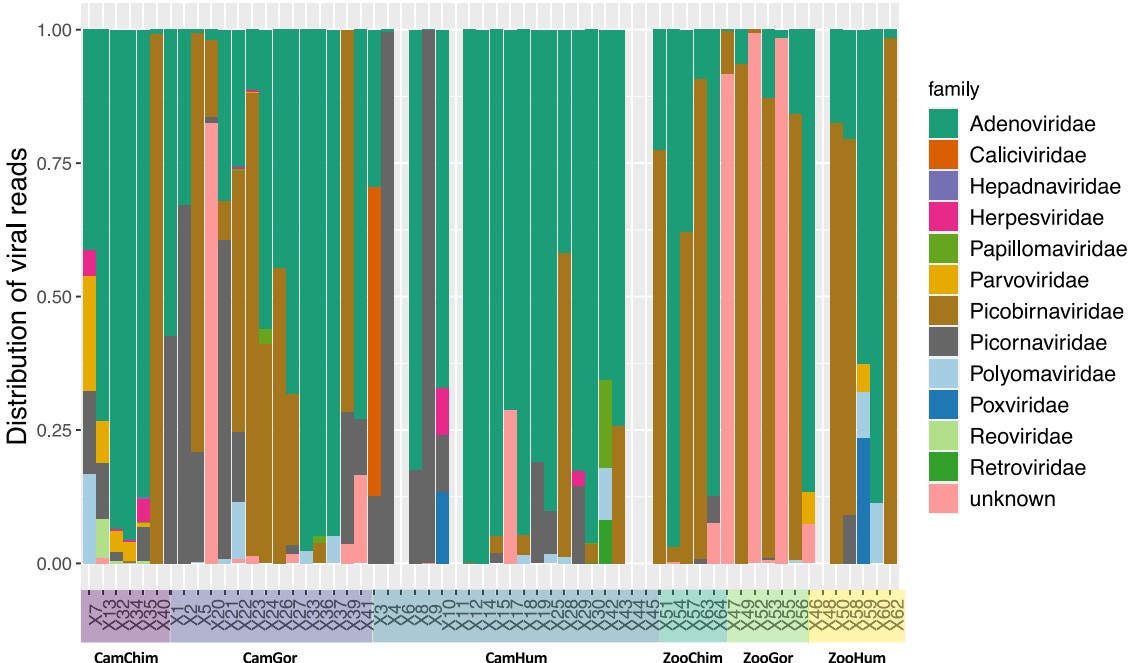

**Fig. 2 | Distribution of viral vertebrate reads per viral family.** Each group is defined by its habitation site (Cam Cameroon, Zoo European Zoo) and animal species (Chimp Chimpanzee, Gor Gorilla, Hum Human). Source data are provided as a Source Data file.

**Table 2 | Number (%) of samples found positive for one of the 10 most prevalent vertebrate viral families by group**

|  | CamChimp n = 6 | CamGor n = 15 | CamHum n = 21 | ZooChimp n = 5 | ZooGor n = 6 | ZooHum n = 7 |
|---|---|---|---|---|---|---|
| Adenoviridae | 6 (100%) | 14 (93%) | 17 (81%) | 5 (100%) | 5 (83%) | 6 (86%) |
| Caliciviridae |  |  | 1 (5%) |  |  |  |
| Hepdnaviridae | 1 (17%) |  |  |  |  |  |
| Herpesviridae | 4 (67%) | 3 (20%) | 2 (10%) |  |  |  |
| Papillomaviridae | 1 (17%) | 2 (13%) | 1 (5%) |  |  |  |
| Parvoviridae | 5 (20%) | 1 (7%) | 2 (10%) |  |  |  |
| Picobirnaviridae | 2 (33%) | 10 (67%) | 6 (29%) | 4 (80%) | 5 (83%) | 3 (43%) |
| Picornaviridae | 4 (67%) | 9 (60%) | 10 (47%) | 2 (40%) | 3 (50%) | 1 (14%) |
| Polyomaviridae | 2 (33%) | 5 (33%) | 5 (23%) |  | 1 (17%) | 2 (29%) |
| Poxviridae |  |  | 1 (5%) |  |  | 1 (14%) |

Each group is defined by its habitation site (*Cam* Cameroon, *Zoo* European Zoo) and animal species (*Chimp* Chimpanzee, *Gor* Gorilla, *Hum* Human).

## Human-great ape contact at the European zoo

Zoo gorillas and chimpanzees lived separately but shared the same environment. Both species had an indoor enclosure with outdoor access on an island with similar vegetation. Islands were surrounded by channels through which water continually circulated. Our observations showed that zookeepers and great apes had notable daily environmental and physical contact. Because zookeepers undertook daily, repetitive tasks that brought them into close proximity (<2 m) with great apes, especially through cage barriers, we did not quantify their contact frequency, but instead observed and documented their activities. Zookeepers handled food during preparation to feed to chimpanzees and gorillas. Moreover, before the COVID-19 pandemic, they generally did not wear masks and gloves when entering the cages each day to remove straw bedding and feces, and when hosing cages with hot water every 3–5 days.

We observed occasional physical contact between zookeepers and chimpanzees (playing, affectionate scratching, grooming), whereas no physical contact occurred between zookeepers and gorillas. Additionally, zookeepers reported that chimpanzees fell ill more often than gorillas, especially with respiratory infections during winter months. Physical contact between zoo great apes and the visiting public, however, is not possible; visitors remain separated from the great apes by a large water channel or by glass walls.

## Fecal sample characterization and comparison

We focused on vertebrate viruses because of their importance for viral transmission and emergence. Among vertebrate viruses, 13 families, 26 genus and 61 species were identified. The most frequent viral families observed in decreasing order were Adenoviridae (53 samples), Picobirnaviridae (30 samples) and Picornaviridae (29 samples). The distribution of viral vertebrates reads per family for each sample tested is shown in Fig. 2. The distribution of viral reads across their natural host categories is depicted in Supplementary Fig. 1.

The number of samples positive for each virus family among the ten most prevalent viral families is detailed in Table 2. Adenoviridae, Picobirnaviridae and Picornaviridae were identified in all human, gorilla and chimpanzee groups and Polymaviridae in all but zoo chimpanzees. Herpesviridae, Papillomaviridae and Parvoviridae were identified only in stool samples of apes and humans in Cameroon.

### Viral diversity and composition

Global virus richness was significantly higher in Cameroon chimpanzees than among Cameroonian humans or zookeepers (Observed and Chao index), although no difference between these same groups was observed with Shannon and Simpson indices (Supplementary Fig. 2a). When we focused only on vertebrate viruses, no significant difference was observed between the six different groups, despite a tendency for higher diversity in Cameroon chimpanzees (Supplementary Fig. 2b).

The comparisons of virome composition using a PERMANOVA with Bray Curtis dissimilarity indices and weighted unifrac showed that the virome differed significantly across different groups (p values < 0.001, Supplementary Table 1). An unsupervised analysis with a PCA among all samples showed that the global virome composition differed between host species, with a closer similarity between zoo chimpanzees and gorillas (Fig. 3a, b). The network projection of virome similarity confirmed the proximity between viromes of zoo great apes (Fig. 3c). Viromes of zoo great apes resembled that of Cameroonian chimpanzees, which in turn was close to that of Cameroonian gorillas. Despite the distinct environments in which stools were collected, the Cameroonian and zoo human viromes closely resembled one another. The network analyses revealed greater virome resemblance between Cameroonian humans and Cameroonian gorillas than between Cameroonian gorillas and other great apes in the Cameroon forest or the zoo (Fig. 3c).

### Viral sharing

We then investigated whether within a shared environment, great apes and humans might harbor viruses with same lowest-common-ancestor (LCA) taxa identified. In Cameroon, a total of 15 vertebrate viral LCA taxa identified in human stool samples were also found in those of chimpanzees or gorillas: 13 viral LCA taxa in gorilla and human stools, 8 in chimpanzee and human stools, and 5 shared by all 3 groups (Fig. 4a). Among these vertebrate viral taxa, the most represented genera were Mastadenovirus (n = 5), Picobirnavirus (n = 3), and Enterovirus (n = 3). In the European zoo, 8 viral LCA taxa were shared between humans and apes, with a greater representation of the Picobirnavirus genus (n = 5) (Fig. 4b).

We analyzed the Mastadenovirus and Enterovirus genera, vertebrate viruses that great apes and humans shared. Concerning the Mastadenovirus genus, we detected adenovirus species D among humans and great apes in Cameroon and in the European zoo. We found species E in Cameroonian humans, gorillas and chimpanzees, and species B in Cameroonian humans and gorillas. Although we detected Adenovirus species B and E in zoo great apes, we did not find them among zookeepers. Enterovirus C and D species were detected only in Cameroonian humans and gorillas.

To evaluate the genetic proximity of the viral sequences, reads belonging to Mastadenovirus and Enterovirus genera were assembled to create contigs. Because the size of the contigs did not permit building of phylogenetic trees to compare different viruses, a network analysis on Cytoscape[31] was performed to assess genetic similarities between shared viruses. The sequence similarity network confirmed the detection of adenovirus species B, D and E (Fig. 5a). The network showed close genetic distances between some adenovirus D collected among humans, chimpanzees and gorillas (Fig. 5a), and between an enterovirus C species collected in one Cameroonian human and those collected from Cameroonian gorillas and chimpanzees (Fig. 5b). The network analysis was further supported for Mastadenovirus in a subset of six individuals by hexon phylogenetic analysis (Supplementary Method 1 and Supplementary Result 1).

### Discussion

To shed light on shared human-great ape viromes and possible mechanisms of bidirectional viral sharing, this study brings together metagenomics analyses, oral histories, anthropological observation, great ape ecological analyses, and real-time evaluation of contact types and frequencies between people and great apes living in the African equatorial rain forest (Fig. 1). In keeping with prior microbiome studies that compare unconfined environments (e.g. forests) and zoo settings, it compares these results to a European zoo, where we easily observed repetitive zookeeper-gorilla and zookeeper-chimpanzee interactions. We had two unexpected findings: the convergence of intestinal virome between Cameroonian humans and gorillas, and the higher proportion of human-

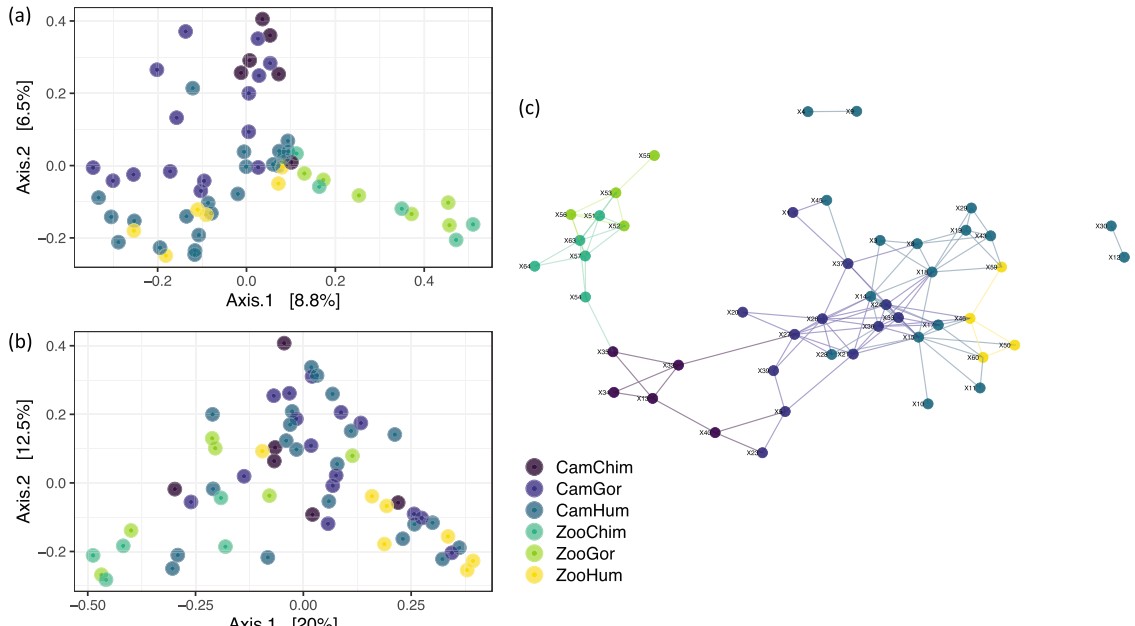

**Fig. 3 | Comparison of intestinal virome composition between humans, gorillas, and chimpanzees in Cameroon and the zoo.** PcoA analyses are based on Bray−Curtis distances (**a**) and on weighted Unifrac distances (**b**). **c** Network plot based on Bray-Curtis distances show similarities among all sample virome profiles.

Only edges connecting individuals (i.e., nodes) with >90% similarity in their virome are shown. Each group is defined by its habitation site (Cam Cameroon, Zoo European Zoo) and animal species (Chimp Chimpanzee, Gor Gorilla, Hum Human). Source data are provided as a Source Data file.

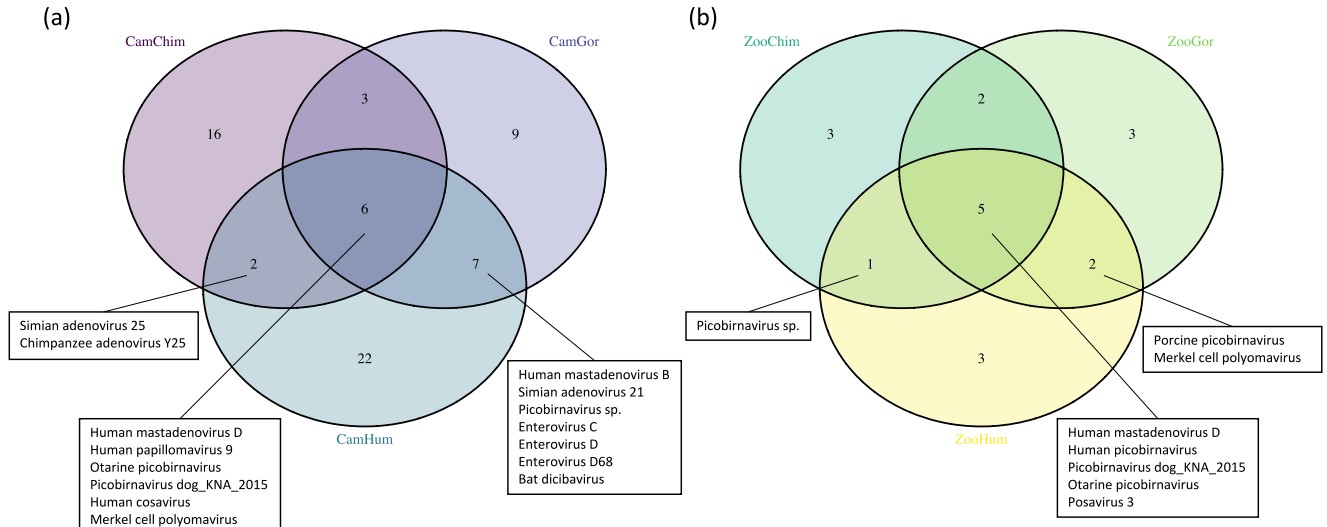

**Fig. 4 | Shared viral taxa between humans, gorillas, and chimpanzees in Cameroon and the European zoo.** Venn diagrams show vertebrate viral lowest-common-ancestor (LCA) taxa in human and great ape stools in Cameroon (**a**) and European zoo (**b**). Each group is defined by its habitation site (Cam Cameroon, Zoo European Zoo) and animal species (Chimp Chimpanzee, Gor Gorilla, Hum Human). Source data are provided as a Source Data file.

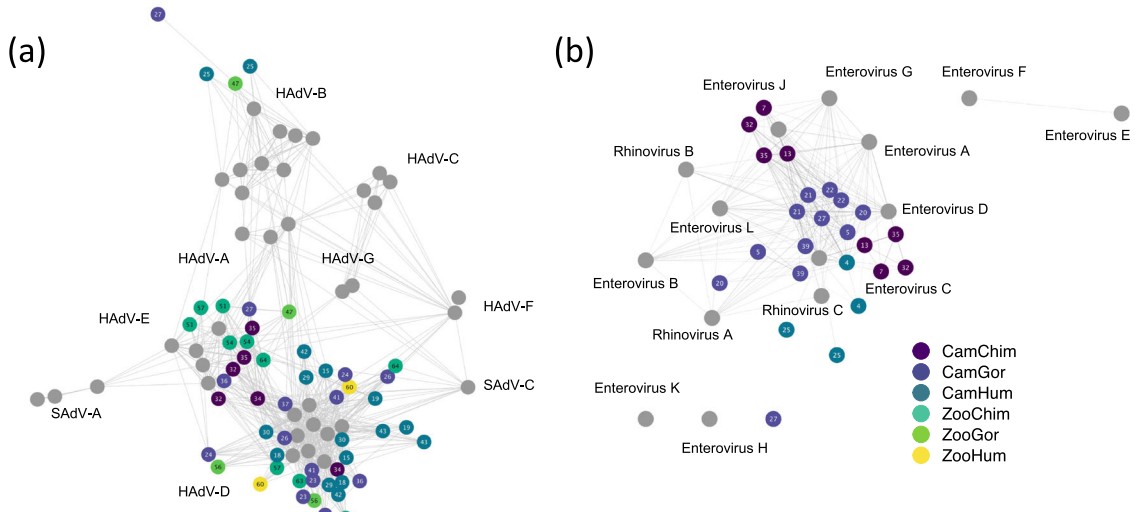

**Fig. 5 | Sequence similarity network representation of Adenoviridae and Picornaviridae contigs.** Networks for Adenoviridae and Picornaviridae are shown in panels **a** and **b**, respectively. Each node represents an individual contig or whole genome reference sequences. Edges are defined based on the Blast Bit-score across individual samples. Each contig is colored according to the group to which it belongs. Gray nodes represent reference sequences. The main metrics on contigs are provided in Supplementary Data 2. Source data are provided as a Source Data file.

great ape sharing in Cameroon. We expected that humans would share more of their intestinal virome with chimpanzees because of their phylogenetic proximity, and that viral sharing would be greater in the zoo, because human zookeepers had closer, more frequent contact with zoo great apes. Additionally, certain adenovirus and enterovirus species were the most frequently shared viruses between Cameroonian great apes and humans. Our multidisciplinary analyses help explain these biological findings in terms of southern Cameroonians' distinct perceptions and practices toward gorillas and chimpanzees, and gorilla and chimpanzee relative densities and behaviors.

The following explains each of our analyses around human-great ape interactions and risks associated with two viral genera, Mastadenovirus and Enterovirus, shared by Cameroonian humans and great apes. We then integrate these analyses to identify complementary mechanisms of human-great ape viral sharing in the

central African forest: physical contact through hunting, meat handling, and fecal exposure; and environmental contact concentrated in small forest gardens, frequented by humans and great apes, especially gorillas.

## A historical framework for human-great ape interactions
Although such evidence is not habitually integrated into metagenomics studies, our qualitative findings shed light on an essential indirect influence on human-great ape interactions and contacts. Southeastern Cameroonians maintain that for millennia, they have shared forest spaces and foods with great apes; historical linguistic and archeological investigations support this claim[32,33]. People now interact less with great apes than in previous generations, but gorillas and chimpanzees still spark avid interest and ambivalence among southeastern Cameroonians. The compelling nature of great apes appears to result from their "charismatic" features; their behaviors lend

themselves to human observation and elicit strong emotional responses among observers[34].

Southeastern Cameroonian mythical tales and historical recollections make no reference to biological evolution. Consistent with historical, anthropological interpretations of mythical tales, southeastern Cameroonians appear to have distilled from their accumulated interactions with great apes their reflections about how humans, gorillas and chimpanzees resemble and differ from one another. These tales comment on an increasing emotional and physical distance between people and great apes, although there is little agreement on how it happened[35–37]. In the absence of long-term great ape population surveys, we cannot conclude that these NHP populations have been depleted. Certain people hunt great apes in this forest, whereas others do not[38].

Our anthropological research shows that southeastern Cameroonians distinguish chimpanzees from gorillas, and such perceptions shape their contacts with these great apes. Chimpanzee behavior, they concur, is closer than gorilla behavior to that of humans. Hence, southeastern Cameroonians contend that they avoid engaging with chimpanzees and are more reluctant to kill and butcher them, a tendency reinforced by chimpanzees' avoidance of humans. Gorillas, by contrast, are reportedly more numerous. Their behavior, perceived by informants as more distant from that of humans, apparently makes it easier for people to kill and butcher them. Human perceptions of these differences between gorillas and chimpanzees seem to yield more frequent human physical and spatial contacts with gorillas than chimpanzees.

Architectural and hygienic features of the European zoo did limit physical contact, but not environmental contact, between human zookeepers and great apes, and as such were part of much longer-term modifications in diets, housing conditions, and separations of captive animals from their keepers, researchers investigating these animals, and visiting publics[39,40]. Nevertheless, even with such limitations, pathogen sharing under contemporary zoo conditions is possible[41].

## More frequent human-gorilla contact than human-chimpanzee contact in Cameroon

These qualitative findings align well with our quantitative analyses of human-great ape contact frequency. Our participatory longitudinal study found elevated, but not significant, human-gorilla spatial contact compared to that of humans and chimpanzees. Our questionnaire dataset, however, revealed significantly more frequent physical contact with gorillas. Most studies assessing human-great ape contact do not evaluate contact frequency[42–44], an important but overlooked factor in understanding pathogenic transmission risks. Our previously published study evaluated quantitative and anthropological data concerning human-great ape physical contact types with diverse NHP species[45]. Here, integrating quantitative contact analyses and qualitative evidence about physical and environmental contacts permits insight into how southeastern Cameroonians may share specific viruses with great apes and closer virome resemblance to that of gorillas.

Our Cameroon investigation also describes the context and locations of these frequent human-great ape physical and environmental contacts. Our multi-village questionnaire showed that great apes were frequently involved in garden crop raiding, and gorillas more so than chimpanzees. Several primate ecological and behavioral factors may explain greater physical and environmental contact with gorillas. First, gorillas appear to be more abundant in the Cameroonian forest, leading to a higher probability of gorillas entering and raiding gardens may explain this phenomenon. Our previous work showed that signs of gorilla presence were 10 times higher than for chimpanzees in this forest[45], in accordance with other studies finding that gorillas are more abundant or in similar densities to sympatric chimpanzees in the Republic of Congo, DR of Congo, Gabon, and Cameroon[46–48].

In addition to different gorilla and chimpanzee relative densities, our southeastern Cameroonian informants also reported that gorillas crop raided more frequently and destructively. For many animals, including gorillas, these gardens were concentrated sites of food raiding and defecation. Gorillas and other nonhuman primates defecated freely in or around gardens. Gardens, frequently distant from villages where people constructed latrines, could also be sites of human defecation during their daily visits. In both cases, defecation could deposit environmentally persistent microbes.

Gorillas and chimpanzees may behave differently in forest gardens, leading gorillas to raid more often than chimpanzees. Most existing literature NHP crop raiding, however, focuses on chimpanzees in east and west Africa[49–52]. Just two studies address mountain gorilla (*Gorilla beringei*) crop raiding[49,52]. Gorilla and chimpanzee behavioral differences in socio-ecological systems require further investigation[53]. Nonetheless, small gardens may be focused sharing sites for environmentally persistent microbes and their exchanges between humans and gorillas.

## Influence of host species and environment on intestinal virome

Our results indicate that each species in its own environment harbored a specific virome composition, as expected. As with gut bacterial microbiota, phylogeny is a strong driver of species-specific enteric virome[20,22,54]. Our findings corroborate a previous investigation of the evolutionary and ecological origins of gut bacteriophage communities (phageome), demonstrating that the phageome structure and dynamics were influenced by superhost phylogeny and environment[55].

Network and dissimilarity analyses revealed that human enteric viromes from Cameroon forest and the zoo were more similar despite habitation in different biotopes, whereas virome composition for great apes appeared to be shaped more significantly by ecology than by species. Additionally, the zoo environment seems to have exercised a greater influence on great ape virome than the forest did for Cameroon great apes. These results corroborate those reported by Moeller and colleagues[56], in which gut microbiota of sympatric chimpanzees and gorillas bore greater resemblance to one another than gut microbiota of either allopatric bonobos and eastern lowland gorillas or allopatric chimpanzees from Tanzania and eastern lowland gorillas. Hence, our findings on human and great ape gut viromes reveal similar patterns to comparative gut microbiomes. Although phylogeny seems to exercise a greater impact on human gut virome and bacteriome, environment apparently has a stronger influence on great ape viromes.

## Targeted assessment of bidirectional viral sharing risks

Among vertebrate viruses known to cause disease in human beings, we identified Mastadenoviruses and Enteroviruses as the major viral genera that humans and great apes share in Cameroon. Both genera can be transmitted through physical and environmental contact.

Mastadenoviruses are associated with many diseases, including mild and severe respiratory infections, gastro-enteritis, encephalitis, cystitis, keratoconjunctivitis and hepatitis. Recently, severe hepatitis cases of unknown etiology among young children have been reported[57]. Because an adenovirus has been detected frequently in these patients' feces or blood, investigators have hypothesized that the etiology of the severe hepatitis is an adenovirus, although investigations are ongoing[57].

Adenovirus is a non-enveloped virus with a double-stranded DNA genome. Primate adenoviruses belong to the genus Mastadenovirus, which includes seven species (A to G) and more than 100 different types. Our analyses found adenovirus species D to be most frequently shared between humans and apes. Adenovirus species D displays frequent recombination events between different types. This tendency to recombine enables the emergence of new types that could escape from previously acquired anti-adenovirus host immunity and potentially trigger disease outbreaks in humans or great apes. Although

adenoviruses are considered highly specific to hosts because of the genome's DNA structure, great ape-human transmission of adenovirus has been shown. A novel adenovirus (TMAdV, titi monkey adenovirus) discovered at the California National Primate Research Center caused a deadly outbreak in a closed colony of New World monkeys (titi monkeys; *Callicebus cupreus*) and infected humans in close contact[58]. Human-to-human transmission of TMAdV was also documented. Other studies have detected specific antibodies of baboon and chimpanzee adenoviruses among caregivers and zookeepers, confirming the capacity of NHP adenoviruses to infect humans[59,60]. Humans may also be capable of transmitting adenoviruses to great apes. Adenoviruses detected among NHPs elsewhere in Cameroon revealed sequences closely related to human adenoviruses[61]. Although no large adenovirus epidemics have been reported, these observations and our findings suggest an elevated risk of cross-species outbreaks.

Recently, machine learning analyzing viral genomes has been used to predict viral zoonotic risk. One study found significantly elevated predicted zoonotic risk in viruses from NHPs and identified adenoviruses among those viruses correlated with the probability of human infection, confirming prior investigations showing that NHP adenoviruses and retroviruses, bat rhabdoviruses, and rodent picornaviruses were more likely to be zoonotic[62,63].

The second most frequent viral genus shared between humans and great apes was enterovirus, known to cause mild to severe disease in humans. Enteroviruses are positive-sense single-stranded RNA viruses with high mutation rates, frequent recombination events, and a potential for newly emerging types. Based on detection of specific antibodies or targeted PCR and specific viral protein sequencing, a few studies provide some evidence for anthropozoonosis and zooanthroponosis[64]. Although enterovirus origins are difficult to ascertain because most have been described in humans, some studies have detected human enteroviruses in zoo-housed and free-ranging NHPs, with variable frequency depending on housing conditions or the degree of cohabitation in urban areas. In urban Bangladesh where NHPs and humans share the same environment, 100% of enteroviruses detected in NHPs were also known to circulate among the human population[65]. In contrast, in a Bangladesh zoo, most Picornaviruses detected in NHPs (53/64; 83%) were simian viruses, with only 8 (12.5%) detected in humans[66]. These results corroborate our findings, suggesting that environmental sharing without specific hygiene control measures enables cross-species transmission of enteroviruses. Another study investigating enterovirus genetic diversity in 615 stool samples collected between 2006 and 2008 from zoo and free-ranging NHPs in Cameroon, the rate of enterovirus detection was 20.2% among zoo NHPs and 3.5% in free-ranging NHPs. These viruses belong to virus types that circulate among humans in 94% of zoo NHP and 55% of free-ranging NHP positive samples[25]. The zoo NHP habitat (large enclosures), where frequent interactions between NHPs and employees and the public, was suspected as a key site facilitating viral transmission.

### Humans, gorillas, intestinal virome, and targeted viruses: convergences in Cameroon

Contrary to our prediction, viromes of Cameroon forest inhabitants and Cameroon gorillas more closely resembled one another than those of zookeepers and zoo gorillas and chimpanzees. The proportion of viral taxa shared between gorillas and humans was four times higher in Cameroon than in the European zoo, and among Cameroon humans and great apes, human-gorilla viral sharing was higher than for human-chimpanzee sharing.

Our historical-anthropological, ecological, and contact analyses suggest two complementary mechanisms to explain viral sharing: physical contact from hunting, meat handling and fecal exposure, and environmental contact through co-use of and fecal exposure in small forest gardens. Support for these mechanisms comes from multiple findings. Perceiving chimpanzees to resemble humans more closely than gorillas, Cameroonian participants avoided chimpanzees and had more frequent physical contact with gorillas through hunting and meat handling, although such activities were not highly frequent. Gorillas appeared more abundant and more active in crop-raiding than chimpanzees in gardens. Small forest gardens thus constituted focused sites of human-gorilla overlap and environmental contact, where people and gorillas could leave behind environmentally persistent enteric viruses.

Human-great ape contact and viral sharing in the European zoo generated unexpected results. Zookeepers and zoo great apes had daily contact and close spatial proximity in a relatively small site, but viral sharing was lower in the zoo than in Cameroon, possibly limited by zookeepers' occasional handwashing, but also by the above-mentioned mechanisms, and notably environmental contact in Cameroonian gardens.

Targeted viral discovery results are consistent with these two sharing mechanisms. Physical and environmental contact can facilitate sharing of Mastadenoviruses and Enteroviruses and could potentially lead to pathogenic spillover or spillback. An increased frequency of physical and environmental contacts between humans and great apes could facilitate the emergence of a novel viral disease in human or NHPs. Social sciences evaluation of human-NHP contact intensity and the introduction of viral surveillance programs where humans and NHPs are in close engagement would be essential for pandemic prevention.

Finally, our investigation reveals the explanatory richness of multidisciplinary investigations of cross-species pathogen sharing. Although limited to correlations, our anthropological-historical and ecological analyses and our granular study of contact type and frequency were essential for explaining the possible routes for viral sharing, illuminated by our metagenomics analyses and targeted viral discovery.

This multidisciplinary study found more human-great ape sharing in Cameroon than in the European zoo, an unanticipated convergence of human and gorilla virome in Cameroon, and apparent sharing of adenovirus and enterovirus taxa. These findings can only be understood by putting into dialogue metagenomics, historical, anthropological, and ecological analyses in southeastern Cameroon. Our analyses point to lengthy human-great ape cohabitation and differential human perceptions of gorillas and chimpanzees, a greater willingness to hunt gorillas, gorillas' higher relative density and greater propensity to raid forest gardens. Interpreted together, our analyses in Cameroon point to two mechanisms facilitating such viral sharing: first, physical contact through great ape hunting, meat handling, and fecal contact, and second, environmental contact via focused co-use of small forest gardens by humans and gorillas.

### Limitations of the study

Our study has several limitations. First, we collected stool samples over one to two weeks, depending on the site, and did not repeat collections. This collection strategy may have influenced our metagenomics analyses. We collected our Cameroon samples during the dry season. Seasonal availability of foods may influence virome composition, and in turn, the similarities and differences observed[67].

Samples were limited in number, primarily because collecting great ape stools in forest settings is challenging. Our samples are, however, numerically sufficient to offer insight into shared gut viromes. We did not collect other NHP stool samples, notably among those with whom Cameroonian people have high-frequency physical or other contact; this investigation focused on great apes because of their greater phylogenetic proximity with humans. Additional sampling would be important for understanding virome sharing between monkeys and human beings in this forest.

Our study investigated the viromes from stool samples and cannot shed light on all potential human-great ape viral transmissions,

notably blood-borne or respiratory viruses. Stool samples may, however, include viruses with a non-digestive tropism. Certain respiratory viruses can be detected in stool samples, such as naked viruses (picornavirus, adenovirus) or some enveloped viruses (such coronaviruses, influenza viruses) but usually with lower viral loads than in the respiratory tract[68]. We found no enveloped respiratory viruses in great apes or humans.

Finally, the European zoo was situated in different ecological conditions from our study site in Cameroon. Nonetheless, multiple microbiome studies have compared unconfined environments and zoo settings[20–24]. In our study, the zoo allowed for daily, easily observed, repetitive interactions between great apes and their zookeepers. Fecal sampling and metagenomic analyses were identical in both two sites.

## Methods

### Study sites and study periods
We conducted research in several villages located between Yokadouma and Moloundou, Cameroon, and at the European zoo. Details of these two study sites are available elsewhere[22]. Team researchers (Authors 1, 7, 8, and 13) made a total of four field trips to Cameroon in 2015, 2016 and 2017 for six months total. We collected data and samples in the European zoo during a five-day research visit in November 2017. To protect the anonymity of human populations and activities, we do not report the names of individual study sites.

### Approvals and authorizations
Ethical approvals for this study were obtained from the Cameroon National Committee for Ethics in Human Research (no. 2015/05/598), the Institut Pasteur Institutional Review Board (no. 2014–30), and the French Committee for the Protection of Persons Ouest V (no. 17/022–2). The Cameroon Ministry of Public Health provided authorization for the research, including all human and great ape sample collection (no. 621_04.16). The Institut Pasteur Committee of Clinical Research also provided authorization for the research in France and Cameroon. We received written informed consent from all participants. We ensured that participant identities could not be ascertained from any data collected (interviews, participatory longitudinal survey, questionnaire, or samples).

### Data collection
**Qualitative data collection.** We began the study by collecting qualitative data using anthropological participant-observation and semi-directed interviews (Supplementary Method 2) with inhabitants of forest villages (Cameroon) and zookeepers (European zoo). In South-eastern Cameroon, we collected 93 in-depth individual and collective interviews with 83 men and 31 women between ages 18 and approximately 85 years, complemented by many informal discussions and 150 h of participant-observation of forest activities and recitation of mythical tales. Gender considerations were crucial in the development and implementation of this method; we sought to determine those gender groups with specialized knowledge and experience with non-human primates and specifically great apes. In compensation for their time, participants received a consumable household good (soap, sugar, salt).

The specific villages in which this research took place are not mentioned for ethical concerns. Illegal hunting and meat trading occurs in certain sites, necessitating the anonymization of these sites to protect informants.

Participant-observations enabled us to observe human interactions with and in proximity to chimpanzees and gorillas. Interviews permitted us to collect qualitative data concerning gorilla and chimpanzee behaviors, diet, habitats, and interspecies contacts, as well as human perceptions and practices with these animals. Participant-observation and semi-structured interviews were conducted in French or in the Bangando language; all interviews were recorded. We collected detailed notes for participant-observations.

In the zoo, Authors 1 and 13 observed and documented zookeeper-great ape interactions and living conditions among the great apes sampled, including feeding regimens, living conditions, and cleaning practices of their habitats. We also observed and conducted nine semi-structured interviews with zookeepers (3 women, 6 men) to document great ape contacts with other animal species, including humans. Because of the daily, repetitive interactions between small numbers of great apes and zookeepers (<10 for each group) within a confined space, we deemed it unnecessary to quantify the number and types of human-great ape contact. Participants received no compensation.

**Participatory longitudinal survey and transects in Cameroon.** Following most of the qualitative data collection, we developed a longitudinal participatory survey for Cameroon volunteers and conducted this survey over 10 months. This participatory longitudinal survey would document human-great ape contacts in real time in the forest, where such contacts were unpredictable. Eighteen volunteers (8 women and 10 men between ages 21 and 63) from the same village collected daily data on their contacts with gorillas and chimpanzees. Again, gender was of central importance in our recruitment and data collection, but we did not achieve full parity in gendered recruitment because the survey required that participants be able to read and write. Although a prior study addressed human-NHP physical contact[45], the present article uses physical and environmental contact data for gorillas and chimpanzees only (Table 1). Because of the time-consuming nature of participation, volunteers received $US 30 per month for their participation in data collection.

**Questionnaire.** Drawing from our preliminary analyses of our qualitative data and participatory longitudinal survey, we developed and implemented a questionnaire (449 participants in four villages, 203 women and 237 men between ages 18 and 83) later in the study to assess human contact frequency with gorillas and chimpanzees. Gender considerations were integrated into the development of this method and in data analyses. Male and female participants were asked if within the last day, week, month, year, or more than one year, they had had specific types of physical and direct (seen alive, heard) contacts with gorillas or chimpanzees. They also were asked to report on great ape involvement in crop raiding, namely the last great ape species raiding one of their fields, as well as the frequency of field raiding.

**Sample collection.** In Cameroon, after identifying great ape nesting sites, we collected fresh stool samples (UBERON: 0001988) during the early morning, taking one stool per nest for a total of 15 gorilla samples and 6 chimpanzee samples. This sample collection commenced several months after the qualitative data collection had begun. At the zoo, we worked with zookeepers to collect stool samples immediately after feces emission, for a total of 6 samples for each species (Gorillas: 1 juvenile female, 2 juvenile males, 2 adult females, 1 adult male; Chimpanzees: 2 subadult females, 4 adult males). These latter collections were conducted after all other field research in Cameroon had been completed. In both sites, fecal sample collection did not attempt to achieve sex parity because of limited numbers of great ape individuals and an overwhelming need to collect one sample only from each individual, but also because collection took place in the forest where parity could not be controlled.

For human stool collection in Cameroon, we included adult participants over 21 years old with no current or chronic health problems. Ethical restrictions necessitated that participants in fecal sampling differed from those in the participatory survey. Participants for both methods lived in the same village. All

participants received an information notice and informed consent form, which were explained orally in the Bangando language. Among potential participants, we conducted a questionnaire to determine whether they suffered from a recent or chronic illness or used medicines (including antibiotics), and what forest-based activities they pursued. All participants received a sampling tube which they used for individual stool collection. All stool samples were delivered within 12 h of emission and were subsequently stored in RNAlater® tubes (Thermo Fisher)[22]. We successfully collected 22 human samples. Participants providing stool samples received a consumable household item (soap or salt) in compensation.

For human stool collection at the European zoo, we invited zoo-keepers working full- or part-time with gorillas or chimpanzees to participate in this study. Stool collection followed the same procedure as in Cameroon, although zookeeper participants stored the sample directly in the RNAlater tube and sent immediately in a secured package the Microbiology Service, Saint Louis Hospital in Paris, France. We have not identified the location of the European zoo to protect the anonymity of the zoo's employees, who are sufficiently small in number and would otherwise be identifiable.

All stool samples were frozen at −80 °C within the 10 days of collection (for Cameroonian samples at Centre Pasteur, Yaoundé, and for zoo samples at Saint Louis Hospital, Paris). The shipment of stool samples from Yaounde to Paris were performed with dry ice without temperature control. Further details of stool collection can be found in a previous publication[22].

A total of 62 stools samples were analyzed, including 29 from humans (22 from Cameroon forest, 7 from European zoo), 21 from gorillas (15 from Cameroon forest, 6 from zoo), and 12 from chimpanzees (6 from Cameroon forest, 6 from zoo). Two samples, one from a Cameroon human (n°38) and another from a zoo chimpanzee (n°61), were removed from final analysis because the sequencing depth was too low (number of total reads <5 million). We recovered a median of 40.6 million of reads per sample (range: 4.11E+06–5.79E+07) for DNA sequencing and of 40.7 million of reads per sample (range: 1.58E+06–1.44E+08) for RNA sequencing. Samples and sequencing information are shown in Supplementary Data 1. The median read per million (RPM) of viral reads per sample was 536 (range 38–97,252, IQR: 137–2367) (Supplementary Table 2).

### Data analyses

**Contact frequency with great apes in Cameroon.** We obtained daily contact data for 18 volunteers over 288 days on average (+/−SD = 21.4, 230−303). We used presence/absence coding of data for one day, one volunteer, one type of contact and one great ape species. We then analyzed the frequency of contact (%) for each type of contacts, each species and each volunteer. We calculated the mean frequency for each type of contacts.

We calculated an estimated frequency of contact (%) using questionnaire data. We assigned a value that was the inverse of the number of days: 100 for daily contact, 14 for once weekly, 3 for once monthly, 0.3 for once yearly and 0.1 for more than once each year (estimated here at once every three years for coding). Based on this scale, we calculated the mean frequency for each type of contact.

For our longitudinal survey and questionnaire datasets, we compared contact frequencies with the two great ape species using a two-sided Mann-Whitney statistical test.

To calculate crop raiding frequency, we analyzed the proportion of respondents who identified a chimpanzee or gorilla as the raiding species for last time their field or garden was pillaged. We also calculated the proportion of respondents citing chimpanzees or gorillas as the species more frequently responsible for crop raiding. Data were coded with Excel 2016 and statistical analyses were performed with R Version 4.2.3[69].

**Qualitative data analyses in both sites.** Following transcription into French of all recorded interviews, we conducted manual coding of all qualitative data (transcriptions and notes), organizing data segments into categories pertaining to descriptions and perceptions of, discrete practices around, and interactions with great apes and other NHPs. From these codes, we used Thematic Analysis to identify broader, cross-cutting themes pertaining to human-great ape engagements, as well as gendered differences and similarities in knowledge and perceptions of great apes. Qualitative analyses took place concurrently with the contact frequency analyses and many months prior to the virome and bioinformatics analysis.

**DNA and RNA virome analysis with shotgun Next Generation Sequencing.** Fecal samples (solid phase) were re-suspended and diluted (50%) in phosphate buffered saline (PBS, ThermoFisher, #10010015) and then centrifuged at 2500 g for 20 min. To enrich for viral particles by reduction of host background, stool supernatant was filtered through a 0.45-μm filter (Corning® Costar® Spin-X® centrifuge tube filters, MERCK, #CLS8162), and an aliquot of 315 μl of filtrate was pretreated before extraction by incubation with different nucleases: TURBO™ DNase (ThermoFisher, #AM2238); Baseline-ZERO™ DNase (Biosearch technologies, #DB0715K); Benzonase® Nuclease (MERCK Millipore, # 70664-3); RNAse One ™ Ribonucleas (Promega, # M4261) for 30 min, at 37 °C. Total nucleic acids were extracted using NucliSENS®easyMAG® (Biomerieux,#280130-280135) according manufacturers protocol. For DNA libraries preparation, 25 μL of extract was used. Depletion of methylated host DNA was performed using NEBNext® Microbiome DNA Enrichment Kit (New England BioLabs, #E2612L) according to the manufacturer instructions. DNA was then purified using DNA Clean & Concentrator® (ZYMO RESEARCH, # D4014) and eluted in 7.5 μL of sterile water. DNA libraries were prepared using Nextera® XT DNA library preparation kit (Illumina, #FC-131-1024). For RNA libraries preparation, Trio RNA-Seq Kit (Nugen, #0506-96) was used according to manufacturer instructions. Libraries were sequenced on an Illumina HiSeq X using 150/150-bp paired-end sequencing.

**Bioinformatics analysis.** Raw reads were cleaned using TRIMMOMATIC[70]. Duplicated reads were removed using Dedupe[71]. Taxonomic assignment was carried out using Kraken2 with Viral, Bacterial and Human Refseq databases[72]. Kraken viral assigned reads were verified using Blastn on Refseq viral database. The e-value cut-off used to assign reads to a particular virus was $10^{-10}$. Reads with inconsistent assignment between Kraken and Blast methods were removed. We conducted a supplemental analysis using Kraken2 with NCBI nt database (Supplementary Method 1), resulting in similar findings which are reported in Supplementary Result 1.

Alpha and Beta Diversity analysis were conducted using packages Phyloseq v1.22.3 and Vegan v2.5–4 in R v3.4.4[73,74]. For Alpha diversity, Simpson, Shannon, Chao1 and Richness index were calculated. For Beta diversity, Bray Curtis dissimilarity and Unifrac metrics were used. Principal coordinate analysis (PCoA) and network analysis was done with either Bray Curtis or weighted Unifrac distance. Permutational analysis of variance (PERMANOVA) was used to compare microbial communities between each group based on Bray Curtis dissimilarity indices, weighted and unweighted Unifrac distances using the adonis2 function with R package vegan.

Viral taxa shared between the human group and at least one great ape group (chimpanzee or gorilla) in Cameroon and in the zoo were identified.

As the genomic coverage of shared viruses between humans and apes did not allow the use of standard phylogeny methods, an alternative approach was used. All viral reads assigned to Mastadenovirus and Picornavirus genera were assembled into contigs with Spade[75]. For each individual, the two largest contigs were selected for further

analysis. We then verified the taxonomic identity of the contigs by performing a blastn on NCBI nt database to check whether they belonged to the viral genera of interest. The main metrics on contigs are given in the Supplementary Data 2. Generated contigs and reference genomes (Supplementary Table 3) were compared using *All versus all BLAST*. Sequence similarity was assessed with the Bit-score and presented through sequence similarity networks with Cytoscape[31].

## Data availability

RNA and DNA Metagenomic raw data are deposited on Sequence Read Archive (PRJNA862314). Qualitative data are not shared to protect anonymity of our respondents. Source data are provided with this paper.

## Code availability

A copy of the analysis code is available at https://doi.org/10.5281/zenodo.7891753.

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

## Acknowledgements

We are grateful to the inhabitants and authorities in southeastern Cameroon for their warm welcome and their support of this study. We also acknowledge the invaluable assistance of the European zoo and its employees where we conducted the investigation. We also greatly appreciate the contributions of Olivia Cheny of the Center for Translational Science at the Institut Pasteur. We thank Marina Vernick for the design of the Fig. 1. The French Agence Nationale de la Recherche (ANR-31-CE31-0004), CIFAR, the INCEPTION project (PIA/ANR-16-CONV-0005), and the National Endowment for the Humanities (US) provided funding for this study (T.G.V.).

## Author contributions

V.N. contributed to study design, data collection, data coding, data analyses and manuscript writing. M.S. contributed to study design, virome analyses and manuscript writing. M.K. and T.H. contributed to data encoding and analyses. A.S.R., S.M.D. and N.R. contributed to virome analyses. S.R. contributed to study design, data collection and qualitative data analyses. P.A. contributed to data collection. R.N. contributed to study design and data collection. F.S. contributed to study design and virome analyses. J.L.G. contributed to study design, virome analyses and manuscript writing. T.G.V. contributed to study design, data collection, data analyses and manuscript writing and editing.

## Competing interests

The authors declare no competing interests.
