## [Peer Review File · Nature Communications]

REVIEWER COMMENTS

Reviewer #1 (Remarks to the Author):

“Unexpected higher convergence of human-great ape enteric viromes in central African forest than in a European zoo: A One Health analysis”. This manuscript integrated different types of data to investigate the differences between humans and apes. It’s an interesting topic but more work is required for this manuscript before further evaluation.

Major points:

1. The Introduction is too lengthy. Please make it shorter (within two pages).
2. The major problem here is the virome data analysis. Kraken2 and RefSeq database were used. RefSeq is actually a very small database for viruses. I assume only a very small part of the data has been analyzed. The reviewer suggests using a bigger database or using assembly-based analysis. And then, the phylogenetic analysis may be able to be processed.
3. Another issue is that phage data were not analyzed. If the authors do not want to include phage data. Please clarify it in the manuscript, and “enteric virome” need to be “enteric eukaryotic virome”. The reviewer suggests including at least one more section for phages.
4. More analyses with details are required. For example, the author collected “contact frequency” data. Can this be integrated into the analysis?

Reviewer #2 (Remarks to the Author):

Overview

The study of microbial transmission between human and nonhuman species is hugely important for both public health and conservation. The authors of the present study seek to improve our understanding of viral transmission between humans and African great apes by supplementing comparative viromic analyses with ethological, ecological, and ethnographic data. They find that the enteric viromes of humans living in Cameroon showed greater similarity to those of sympatric gorillas than did the viromes of zookeepers and gorillas living in a European zoo. They also found

that Cameroonians' viromes were more similar to sympatric gorillas' than to chimpanzees', which accords with ethnographic data suggesting a history of frequent interactions between humans and gorillas. The authors conclude that shared space and physical interactions are, in this case, more important in shaping the virome than is genetic proximity.

This is a very interesting study that uses a commendably multi-tiered dataset. I do, however, have some concerns and comments for the authors, as listed below:

Major Comments

Lines 92-123: The ethnographic data concerning Cameroonians' perceptions of chimpanzees and gorillas are fascinating. I would argue, however, that they are not always clearly or meaningfully connected with your study's overarching purpose or main findings. I would recommend that the ethnographic data be more succinctly summarized, moved to supplementary material, or far more explicitly tied into the virome results.

Lines 189-200: Do these analyses include all viruses or only vertebrate viruses? It's not clear to me, and this is important.

Lines 262-270: In contrast to the following paragraph, the ethnographic data here seem tangential to your study. More specifically, the information here does not clearly contribute to your argument that Cameroonians interact with chimpanzees less often than gorillas. Given that you are integrating a wide range of data, I would urge you to be more concise and retain only those data that directly contribute to your theses.

Line 470: It's not clear to me when the qualitative data were collected relative to the quantitative data. Were they collected during the same trips? Most importantly, were the qualitative (ethnographic) data collected after and in response to the quantitative data, so as to shed more light on your virome results?

Lines 472-473: The almost complete absence of information about study sites is a problem. The preservation of individual anonymity does not require you to be quite so vague. (I'm particularly concerned that the name of the zoo is not even reported, which seems to me an atypical approach.) This problem is compounded by the authors' decision not to make their datasets publicly available (with the exception of sequence reads). Consequently, I have very little way of confirming or judging the veracity of the reported results.

Lines 583-585: What was the percent similarity cut-off used to assign reads to a particular virus?

Minor Comments

Line 15: The word “explain” is a little strong. Your results largely hinge on self-reported contact rates. More data on observed contact rates, spatial use by sympatric gorillas and chimpanzees, and environmental sampling of viruses—in particular, sampling from Cameroonian gardens visited by gorillas—will help make your arguments stick. (I am not suggesting that more data be collected for this study but that you should soften your language.)

Line 47: Be careful with the word “holistic”. Although you are integrating a truly admirable variety of data, there are important pieces of the puzzle that your study does not touch – host immunity and physical condition, to name only two examples. “Integrative” is more appropriate.

Discussion: I am curious what the authors make of the processing of food items for the zoo-housed apes. Are food items prepared in such a way that the transmission of viruses via food items is highly unlikely?

Table 1: I recommend that you clarify in the caption what exactly you’ve compared using Mann-Whitney tests (i.e., what is the difference that’s being tested?).

Reviewer #3 (Remarks to the Author):

Interdisciplinary research is needed to understand the drivers of zoonotic transmission of pathogens between human and non-human animals. This study is exemplary of an interdisciplinary approach of contact between human and non-human primates in Cameroon. It uses ethnographic approaches to understand the microbiological findings on the shared viromes between humans, gorillas and chimpanzees. Its main finding is that gorillas share more virome with humans than chimpanzees. Because gorillas are phylogenetically more distanced from humans than chimpanzees, humans are less reluctant to hunt them, which causes high viral sharing through meat cutting. Gorillas also share human habitats by entering small gardens where they leave feces. This article thus shows that environmental relations are a stronger cause of viral sharing than phylogenetic proximity. This a very

strong finding with very clear evidence. The limits of the collect of samples in Cameroon are acknowledged, and the method could be repeated so that the findings can be confirmed.

A positive control in a European zoo is mentioned which I find less convincing. The hypothesis of the authors is that the convergence of viromes between humans and great apes in the zoo would be higher because of frequent contacts. But the viromes turns out to be less convergent than in Cameroon villages, which the authors explain by the more frequent hand washing. However very little is said about the contacts between zookeepers, gorillas and zoos : the location of the zoo is not given (why not give at least its country, since "Cameroon" is mentioned for the main study and not "Africa") and no reference is given to works on zoonoses in zoos in biology, history, anthropology of history (this is a strong field of research in environmental history, e.g. in France the works of Eric Baraty and Violette Pouillard). Have contacts between humans and great apes in zoos changed over the last century ? It would be necessary to symetrize the accuracy of analysis in Cameroonian villages and the "European zoos" to make it a real positive control reinforcing the conclusion on the environmental drivers of zoonotic pathogens.

Response to reviewers

We thank the three reviewers for their careful evaluations of the manuscript. Below is our line-by-line response to each criticism and recommendation.

Reviewer 1:

1. *The Introduction is too lengthy. Please make it shorter (within two pages).*

We recognize that the length may concern some readers and have therefore cut some material from the introduction. It is now just slightly over 2 pages. Further cutting beyond its current state, however, would entail omitting crucial material that could hamper reader understanding of the manuscript. This manuscript is unusual in that it brings into dialogue three different types of evidence – metagenomic, ecological, and anthropological. It thus requires additional initial explanation for readers to understand what we are doing and why.

2. *The major problem here is the virome data analysis. Kraken2 and RefSeq database were used. RefSeq is actually a very small database for viruses. I assume only a very small part of the data has been analyzed. The reviewer suggests using a bigger database or using assembly-based analysis. And then, the phylogenetic analysis may be able to be processed.*

We agree with the reviewer that the RefSeq database is small. Nevertheless, we opted to use a conservative strategy to analyze the virome data. The main objective of the study was to determine eukaryotic viruses sharing between humans and non-human primates, and therefore, our conservative and careful analytical strategy was to rely on specific data provided by **curated databases such as RefSeq**. Although non-curated larger databases might be of interest to identify unknown viruses in a specific context (e.g. an infectious syndrome without any etiology), such databases are less relevant for comparing viromes between species because false assignments occur frequently.

We used an assembly-based analysis by using contigs to assess genetic similarities between the main shared viruses (Adenovirus and Enterovirus) because the genomic coverage did not allow the use of standard phylogeny methods. The use of a larger, non-curated database might enable us to detect a larger number of potential viral reads **but does not allow us to increase the size of contigs**.

Following Reviewer 1's recommendations, however, we did conduct a reanalysis of our virome data by using Kraken2 with the NCBI nt/nr database. We present below and in a Supplementary file of the resubmitted manuscript the main results of this new analysis. Overall, we would signal that the main findings of virome sharing between humans and great apes closely resemble our original analysis.

- ➔ The use of NCBI nt/nr database increased significantly the alpha diversity for all viruses and vertebrate viruses. Although differences in alpha diversity of viromes with all viruses between great apes and humans from the zoo or Cameroonian forest were observed, no difference of alpha diversity for vertebrate viruses were found using the larger database (Figure 1). The differences in alpha diversity for viromes including all viruses may depend on bacteriophages and invertebrate viruses. As recently published by different groups, further

work is required to expand virus databases with correct classification, in particular for bacteriophages and invertebrate viruses (Expansion of the global RNA virome reveals diverse clades of bacteriophages. Neri U et al. Cell. 2022 Oct 13;185(21):4023-4037.e18. doi: 10.1016/j.cell.2022.08.023; Hyperexpansion of RNA bacteriophage diversity. S.R. Krishnamurthy et al. PLoS Biol., 14 (2016), p. e1002409)

→ The comparisons of virome composition using a PERMANOVA with Bray Curtis dissimilarity indices and weighted unifrac showed that ***similar to our original analysis, the viromes differed significantly across different groups*** (p values < 0.001).

→ The network projection and PCoA of virome similarity confirmed the results obtained with RefSeq databases. Viromes of zoo great apes resembled that of Cameroonian chimpanzees, which in turn was close to that of Cameroonian gorillas. Cameroonian and zoo human viromes closely resembled one another. A greater virome resemblance between Cameroonian humans and Cameroonian gorillas than between Cameroonian gorillas and other great apes in the Cameroon forest or the zoo was also found. ***These results closely resemble our original analysis.***

→ Viral sharing

The investigation of viral sharing within a shared environment between great apes and humans was assessed with same lowest-common-ancestor (LCA) taxa identified. The new analysis identified more vertebrate virus families, 40 versus the 13 found through the first analysis. However, several families were present in each species in each environment and for almost all individuals, ***suggesting likely taxonomic assignments errors related to incorrectly referenced sequences in NCBI nt/nr database*** (Table 1, Supplementary data). In addition, the analysis with NCBI nt/nr identified some viruses very unlikely to be detected, such Arenaviridae in zookeepers, and bovine fever ephemerovirus, a rhabdovirus detected in humans and great apes in both environments. ***We consider these results to be poorly specified, and thus unable to reflect reliable sharing between great apes and humans.***

Detailed viral sharing is illustrated in the Venn diagrams, with a list of viruses for each human–great ape (chimpanzee or gorilla) intersection. ***Similar to our previous analysis with RefSeq, we also found a higher proportion of sharing in Cameroon between humans and gorillas.*** The number of viral taxa shared between gorillas and humans was also higher in Cameroon than in the European zoo. Among Cameroon humans and great apes, human-gorilla viral sharing was higher than for human-chimpanzee sharing with the two analyses (13 vs 8 with RefSeq and 91 versus 44 with NCBI nt/nr).

Regardless of the database used, families most frequently shared remained the same: adenoviridae, picornaviridae, and picobirnaviridae. To evaluate the genetic proximity of the viral sequences, we used the same approach with contigs. ***Our results from the first and second analyses are identical.***

Given these results in the second analysis and their close resemblance to the first, we therefore suggest retaining our primary analysis in the main text and putting the complementary analysis using NCBI nt/nr in our Supplementary data. We will also signal that we have conducted this second analysis in our Methods section (Lines 633-634).

3. *Another issue is that phage data were not analyzed. If the authors do not want to include phage data. Please clarify it in the manuscript, and “enteric virome” need to be “enteric eukaryotic virome”. The reviewer suggests including at least one more section for phages.*

The objective of our work was to assess the eukaryotic virome sharing, because eukaryotic viruses are the only ones that could be associated with emerging viral disease. In response to this recommendation, we have altered our text to specify “enteric eukaryotic virome” in the title, abstract and throughout the manuscript itself.

4. *More analyses with details are required. For example, the author collected “contact frequency” data. Can this be integrated into the analysis?*

This contact frequency data is already included in the initial draft. Because of ethical restrictions, human participants who reported on their contact frequency (“volunteers”) and those sampled for fecal analysis were not the same individuals. They lived, however, in the same village. We have clarified this point in the text (Lines 557-559).

Reviewer 2

1. *The ethnographic data...are not always clearly or meaningfully connected with your study’s overarching purpose or main findings. I would recommend that the ethnographic data be more succinctly summarized, moved to supplementary material, or far more explicitly tied into the virome results.*

We thank the reviewer for this observation and take very seriously the importance of integrating the ethnographic data into our overall analysis. Because the historical and ethnographic data and analysis are essential, we have opted for the third option (linking more explicitly to the virome results, although we have also pared down some of this text. To that end, we have added explanations in Lines 96-98, 139-143, 274-276. Most important, we have **created an additional Figure 1** (first introduced in Line 92; referred to in Line 253) to emphasize clearly how these different analyses are tied to the virome results.

2. *Lines 189-200: Do these analyses include all viruses or only vertebrate viruses? It’s not clear to me, and this is important.*

Our analyses include only vertebrate viruses, and in response to this question, we have specified this point in the text (Lines 191, 192, 206).

3. *Lines 262-270: In contrast to the following paragraph, the ethnographic data here seem tangential to your study. More specifically, the information here does not clearly contribute to your argument that Cameroonians interact with chimpanzees less often than gorillas. Given that you are integrating a wide range of data, I would urge you to be more concise and retain only those data that directly contribute to your theses.*

Again, we have more explicitly made this linkage between ethnographic data and our central question through the inclusion of a new Figure 1 and in the text (Lines 96-98, 139-143, 274-

276). These results are important because they indirectly increase the willingness and frequency of human contact with gorillas in comparison with chimpanzees.

4. Line 470: It's not clear to me when the qualitative data were collected relative to the quantitative data. Were they collected during the same trips? Most importantly, were the qualitative (ethnographic) data collected after and in response to the quantitative data, so as to shed more light on your virome results?

The reviewer raises crucial questions here, and in response, we have clarified the order of data collection in the Methods section (Lines 503, 528, 539-40, 549-550). Briefly, the qualitative data were collected BEFORE the quantitative data AND the stool collection and virome analyses. Indeed, we used initial analyses in the qualitative data to strengthen our quantitative tools, effectively to include more precise questions pertaining to actual practices in our survey. The virome analysis occurred after all other data were collected and analyzed. We could contend that the order of our multidisciplinary data collection and analysis is a great strength of the study because we didn't know until we put all of our distinct disciplinary analyses together that they in fact complemented one another in important ways.

5. Lines 472-473: The almost complete absence of information about study sites is a problem. The preservation of individual anonymity does not require you to be quite so vague. (I'm particularly concerned that the name of the zoo is not even reported, which seems to me an atypical approach.) This problem is compounded by the authors' decision not to make their datasets publicly available (with the exception of sequence reads). Consequently, I have very little way of confirming or judging the veracity of the reported results.

We recognize the reviewer's concerns about transparency here, but we are seriously ethically constrained by several factors. First, the European zoo explicitly insisted that we *not* provide its name or identify its location in any reports or publications, because if zookeepers and zoo gorillas and chimpanzees were found to share any potential pathogens, zoo administrators would face severe consequences from state veterinary authorities. We are obliged to respect their wishes.

In terms of the village names where we worked, we identified only the region (southeastern Cameroon, between Moloundou and Yokadouma). But because hunters and trappers are sometimes engaged in illegal hunting activities, and because for its part the state has undertaken punitive measures against even inhabitants who do respect hunting laws, we have not identified specific villages to protect our informants.

We have made available sequence reads, transect data, and contact data. We do not, however, share interviews or participant-observation notes because our ethical approvals from Cameroon and French explicitly prevent us from doing so. Qualitative data is sensitive, and the specific aspects of certain types of specialized knowledge can potentially permit identification of the source of data.

We note these constraints in the Materials and Methods section, Lines 512-514, 571-573.

6. Lines 583-585: What was the percent similarity cut-off used to assign reads to a particular virus?

The e-value cut-off used to assign reads to a particular virus was 10^{-10} . This information has been added in the Materials and Methods section (line 632).

Minor Comments

Line 15: The word “explain” is a little strong. Your results largely hinge on self-reported contact rates. More data on observed contact rates, spatial use by sympatric gorillas and chimpanzees, and environmental sampling of viruses—in particular, sampling from Cameroonian gardens visited by gorillas—will help make your arguments stick. (I am not suggesting that more data be collected for this study but that you should soften your language.)

Reviewer 2’s point is well taken. In response, we have softened the language in multiple places in the text (“helps explain”, “suggests”) (see for example Lines 15, 52, 426).

Although it is true that our results include self-reported contacts, the granularity and real-time collection of this data supports its reliability. In addition, we present rigorously collected and analyzed anthropological data from experienced field anthropologists, transect data and survey data to support our claims.

Line 47: Be careful with the word “holistic”. Although you are integrating a truly admirable variety of data, there are important pieces of the puzzle that your study does not touch – host immunity and physical condition, to name only two examples. “Integrative” is more appropriate.

We agree and have thus have altered the term to “integrative” to reflect better what we are doing (Lines 8, 49).

Discussion: I am curious what the authors make of the processing of food items for the zoo-housed apes. Are food items prepared in such a way that the transmission of viruses via food items is highly unlikely?

We have added additional line that food items are washed prior to feeding. (Line 179-180).

Table 1: I recommend that you clarify in the caption what exactly you’ve compared using Mann-Whitney tests (i.e., what is the difference that’s being tested?).

We thank the reviewer for pointing out this lack of clarity and have have added a clarification in the caption of Table 1.

- *A positive control in a European zoo is mentioned which I find less convincing. The hypothesis of the authors is that the convergence of viromes between humans and great apes in the zoo would be higher because of frequent contacts. But the viromes turns out to be less convergent than in Cameroon villages, which the authors explain by the more frequent hand washing.*

No control, particularly in field settings, can control for all conditions. Still, we were able to observe easily all forms of contact (physical, direct, indirect, and spatial) between zookeepers and zoo great apes; we have published a full description of zoo conditions in a previous publication (Narat et al. Sci Rep 2020) and have cited it in the present manuscript; and we have cited secondary literature documenting pathogen sharing between zookeepers and great apes in a zoo. Thus, we would contend that the zoo does fulfill the criteria of a control. (Lines 70-71, 253, 258, 436, 474-477, 486)

- *However very little is said about the contacts between zookeepers, gorillas and zoos: the location of the zoo is not given (why not give at least its country, since "Cameroon" is mentioned for the main study and not "Africa") and no reference is given to works on zoonoses in zoos in biology, history, anthropology of history (this is a strong field of research in environmental history, e.g. in France the works of Eric Baraty and Violette Pouillard). Have contacts between humans and great apes in zoos changed over the last century? It would be necessary to symmetrize the accuracy of analysis in Cameroonian villages and the "European zoos" to make it a real positive control reinforcing the conclusion on the environmental drivers of zoonotic pathogens.*

As we indicated in our response to Reviewer 2, we cannot provide any identifying information concerning the zoo, including the country where it is located. The problem is that the number of zoos in Europe is already limited, and providing our description (including zoo architecture), the presence of these specific great apes, and country would immediately offer readers sufficient evidence to identify the zoo where we conducted this research.

We greatly appreciate the reviewer's pointing out the relevance of the history of zoos in Europe and changing contact between zoo great apes and humans. We have therefore added additional lines in the results to reflect better the zoo's architectural structure that prevent contact (Lines 186-188) and in the discussion (Lines 301-305), providing some historical context for the architectural and hygienic separations between zoo great apes and their zookeepers. In addition, we have added a reference to one work documenting pathogen sharing in one zoo, although we are required to limit our references to stay within *Nature Communications* guidelines.

REVIEWER COMMENTS

Reviewer #1 (Remarks to the Author):

Thanks for the responses from the authors. The manuscript improved significantly, however, the data analysis needs to be more comprehensive. I can understand some information cannot be shared with the public, but more interpretations are required for the sequencing data analysis.

1. The reviewer appreciates that the author added analysis using Kraken nt/nr database. Actually, nt and nr are two databases. Could the authors clarify which one did they use? This analysis requires a lot of disk space and RAM. Could the authors acknowledge the computing resource in the manuscript?
2. In the "response letter", the author claimed, "a larger number of potential viral reads does not allow us to increase the size of contigs". This isn't very clear. In the Method, the author assembled contigs based on reads assigned to the targeted taxa. If you have more mapped reads, you should be able to get better contigs.
3. More data are required in the "Viral sharing" section. Figure 4 is fine. In Figure 5, the readers may want to know 1)how many reads were used to build contigs; 2)what are the contig lengths; 3)did the author align the contigs back to the database to check whether they are Adenovirus/Enterovirus or not; 4)are the network data convincing enough to support the taxonomic assignments of these contigs; 5)can the authors use protein similarity (hexon, for example) for the phylogenetic analysis. A deep phylogenetic analysis here will contribute a lot to this part.

Reviewer #2 (Remarks to the Author):

I appreciate the authors' effort in responding to reviewer questions and concerns. My comments on the original draft have been appropriately addressed. I have only two remaining comments.

1. I thank the authors for explaining why they have provided limited identifying information about the zoo in which they conducted this work. Their response highlights what strikes me as a troubling ethical issue—that a zoo would fear and potentially dissuade proper investigation of microbial transmission between humans and nonhuman animals, especially given what an important research area this is (and that, for instance, transmission of SARS-CoV-2 has been observed at zoos).

2. To echo Reviewer 3's comments, I would encourage the authors to avoid using the word "control" in reference to the zoo. To my mind, the term implies that precise causal mechanisms about human-ape viral transmission can be extrapolated based on differences in the European and Cameroonian settings. Given the sheer number of differences between these two settings that may plausibly explain differences in viral transmission, I suggest that the authors find another way to describe their research design that does not rely on terms like "control" and "treatment".

Reviewer #3 (Remarks to the Author):

I am satisfied with the remarks on the absence of physical contacts between zookeepers and primates and about the necessity to conceal the location of the European zoo.

We have revised the manuscript, following the questions and recommendations of Reviewers 1 and 2, and we thank them both again for their careful and thoughtful reading of the revised manuscript. We note that we have responded to all questions and made the necessary changes.

1. Reviewer 1

Thanks for the responses from the authors. The manuscript improved significantly, however, the data analysis needs to be more comprehensive. I can understand some information cannot be shared with the public, but more interpretations are required for the sequencing data analysis.

We thank Reviewer 1 for these comments and provide further interpretation below in our response.

The reviewer appreciates that the author added analysis using Kraken nt/nr database. Actually, nt and nr are two databases. Could the authors clarify which one did they use? This analysis requires a lot of disk space and RAM. Could the authors acknowledge the computing resource in the manuscript?

We thank the reviewer for pointing out this problem. We used the database nt and have made the necessary correction in the supplemental file.

Please note that we have uploaded an author change form signed by all authors to add Abdeljalil Senhaji Rachik as co-author in acknowledgement of his conduct of the bioinformatics analysis. We have also expressed our thanks to the Institut Français de Bio-informatique (IFB) for having provided us with the necessary computing resources for our analyses.

In the "response letter", the author claimed, "a larger number of potential viral reads does not allow us to increase the size of contigs". This isn't very clear. In the Method, the author assembled contigs based on reads assigned to the targeted taxa. If you have more mapped reads, you should be able to get better contigs.

We have expressed ourselves unclearly and again, thank the Reviewer for this criticism. For some individuals, the size or the number of contigs could be increased, but few individuals had contigs of good quality that covered regions of interest to perform the phylogenetic analysis. We explain this in detail in our following points below.

More data are required in the "Viral sharing" section. Figure 4 is fine. In Figure 5, the readers may want to know 1) how many reads were used to build contigs; 2) what are the contig lengths; 3) did the author align the contigs back to the database to check whether they are Adenovirus/Enterovirus or not; 4) are the network data convincing enough to support the taxonomic assignments of these contigs; 5) can the authors use protein similarity (hexon, for example) for the phylogenetic analysis. A deep phylogenetic analysis here will contribute a lot to this part.

We used all reads assigned as belonging to Adenoviridae and Picornaviridae by our analysis pipeline. We then verified the taxonomic identity of the contigs by performing a blastn on NCBI nt to check whether they belonged to the viral families of interest. Following the reviewer's advice, we added additional information in the Methods (Lines 644-645, 656-665), as well as an additional table in the Supplementary Table 3 with the lengths and the number of reads for each contig used in the network analysis and the taxonomic identification found by blastn on NCBI nt.

Following the Reviewer's advice, we regenerated contigs from reads assigned as Adenoviridae during the second analysis on an extended database (NCBI nt). We then annotated each contig generated from a database of adenovirus proteins and selected the contigs corresponding to the hexon protein. We were thus able to isolate hexon contigs for 6 individuals, which we aligned with Adenovirus reference sequences to generate a phylogenetic tree. Although the number of individuals analyzed is low, this tree confirms the results found with the network. Individual 47 (ZooGor) is found infected with HadV of species B in both analyses. Individuals 54 and 51 (ZooCHim) also were infected with HadV-E in both analyses. Finally, concerning Cameroon, we can observe in the tree that for two humans from Cameroon (individuals 12 and 30), we found HAdV D, and that HAdV from a Cameroon gorilla (individual 36) was phylogenetically close (bootstrap 100%) to that of one of these Cameroonian humans (individual 12). **These results thus confirm those found by the network analysis.** We have added the phylogenetic tree based on the hexon protein coding region in the supplemental material. Concerning Picornaviridae, we unfortunately did not have enough contigs covering regions of interest (such as VP1 or VP2/VP4) to perform the same analysis.

We added a figure in the supplementary analysis, and we added a text in the paragraph concerning viral sharing in the results section (Line 227).

2. Reviewer 2

Reviewer #2 (Remarks to the Author): I appreciate the authors' effort in responding to reviewer questions and concerns. My comments on the original draft have been appropriately addressed. I have only two remaining comments.

I thank the authors for explaining why they have provided limited identifying information about the zoo in which they conducted this work. Their response highlights what strikes me as a troubling ethical issue—that a zoo would fear and potentially dissuade proper investigation of microbial transmission between humans and nonhuman animals, especially given what an important research area this is (and that, for instance, transmission of SARS-CoV-2 has been observed at zoos).

We agree with Reviewer 2 that the non-identification of the zoo site raises an ethical question: were the study to identify a pathogen shared between zoo great apes and zookeepers, would it not be necessary to identify this zoo?

Two considerations far outweigh this ethical question. First, identifying the zoo would permit the identification of zookeeper participants in our study, because the number of zookeepers caring for great apes is very limited there. Divulging the zoo identity would therefore put us in **violation of the respect for the privacy and confidentiality of our human subjects at the zoo. This respect for confidentiality is an overriding principle of national, European and international ethical standards concerning the protection of human subjects in research, enshrined in the Declaration of Helsinki (1964) and the Belmont Report (1976).** A national ethics committee reviewed and approved our study, and we must adhere to the standards that it sets.

Second, to address the possibility that our study yielded findings that indicated a serious risk for zookeepers and/or zoo great apes, we remained in continual contact with the zoo to inform them of our results. We did not identify a serious pathogen shared between zookeepers and great apes. Had we found such a result, we would have immediately informed the zoo, insisting that they alert country health authorities. Had they failed to do so, we would have contacted these authorities. Hence,

although we will not identify of the zoo *in a publication*, we assiduously respect the need to inform necessary actors and authorities – and were prepared to do so if our study had revealed any findings that would endanger zookeepers, zoo great apes, and the public.

Hence, although our non-identification of the zoo raises a question about transparency that we take very seriously, publishing the name of the zoo would entail a serious ethical violation. We put into place measures to mitigate the ethical question raised by Reviewer 2.

3. Reviewer 2

To echo Reviewer 3's comments, I would encourage the authors to avoid using the word "control" in reference to the zoo. To my mind, the term implies that precise causal mechanisms about human-ape viral transmission can be extrapolated based on differences in the European and Cameroonian settings. Given the sheer number of differences between these two settings that may plausibly explain differences in viral transmission, I suggest that the authors find another way to describe their research design that does not rely on terms like "control" and "treatment".

In response to this remark, we have removed the term "control" altogether from the manuscript but retain the comparison between the two sites. Instead, we refer to the forest site and a "compared environment", the zoo. We never use the term "treatment".

Nevertheless, we have retained the comparison between the forest and zoo on the basis of the following justification:

Comparing the microbiome of animals in zoos and natural settings **is a widespread, fully accepted practice in peer-reviewed microbiome studies. Such comparisons identify the effects of environment on microbiome composition** (see, for instance, Campbell et al, *ISME* 2020; Hale et al, *Microb Ecol* 2018; MacKenzie et al. *Integr J Microbiol*, 2017; Amato et al *Oecologia* 2016; Bennett et al. *Am J Primat* 2016; Moeller et al. *PNAS* 2014; Barelli et al. *Sci Rep* 2015; McCord et al. *Am J Primat*, 2014, and many others). Such comparisons draw from a long-established and accepted practice of comparison across sites in ecological studies. None of these studies employ the term "control", but rather justify their comparisons between zoos and natural settings because the former offer simpler environments with smaller numbers of investigated animal species, easily observed conditions and contacts with human beings, and predictable diets. These real-life comparisons are essential: without them, we would be deprived of crucial scientific knowledge about the effects of environmental (including diet) differences on gut microbiome composition.

Our investigation aligns with this widely accepted practice in microbiome (and ecological) studies. We employed **identical sample collection methods and next generation sequencing and bioinformatic analyses of human and great ape viromes and viral sharing** in a zoo and the forest. In keeping with these microbiome studies, we employed the zoo as a comparison because it offered a simpler environment than the forest, with fewer great apes and fewer people (zookeepers) in contact with them. Engagements between zookeepers and great apes were highly visible, easily observed, and highly repetitive, in contrast to those of humans and great apes in the forest.

Where our study departs from conventional microbiome studies – and constitutes a major contribution – is that **we fully integrate anthropological and ecological data and analysis with our virome and viral sharing analyses**. That Reviewer 2 found the two study sites "different" results directly from the fact that we take seriously the socio-cultural and ecological relations and

features that may affect human-great ape viral sharing. Such features often go unexplored in studies of microbial sharing across species, which instead rely on stereotyped explanations (e.g. deforestation and hunting) without reference to actual human practices and ecologies.

To allay concerns of all readers, we have integrated the above explanations into the manuscript (Lines 56-57, 156-160, 241-243, 461-465).

We take seriously the concerns of the Reviewers, whose recommendations have generated considerable discussion within our team and raise important disciplinary questions, ones that we have addressed in our publications about collaboration and causation across the disciplines in microbiome studies (Amato et al. *Bioessays*, 2019) and zoonotic spillover (Narat et al. 2017). In addition to its contributions to multi-disciplinary investigation of spillover and spillback, this manuscript can catalyze further reflection across multiple biomedical and social sciences disciplines, yielding innovative research and new knowledge.

Reviewer #3 (Remarks to the Author): I am satisfied with the remarks on the absence of physical contacts between zookeepers and primates and about the necessity to conceal the location of the European zoo.

We sincerely thank this Reviewer for the helpful remarks.

REVIEWERS' COMMENTS

Reviewer #1 (Remarks to the Author):

All my comments have been addressed.

Reviewer #2 (Remarks to the Author):

I thank the authors for seriously engaging with the concerns raised by the reviewers. My questions and comments have all been satisfied. This is an excellent study.

Responses to REVIEWERS' COMMENTS

Reviewer #1 (Remarks to the Author):

All my comments have been addressed.

We sincerely thank Reviewer 1 for the constructive criticisms and comments over the course of multiple reviews.

Reviewer #2 (Remarks to the Author):

I thank the authors for seriously engaging with the concerns raised by the reviewers. My questions and comments have all been satisfied. This is an excellent study.

We are deeply appreciative of Reviewer 2's full engagement with the multiple iterations of this manuscript. Their concerns and suggestions have strengthened the piece.